

# Time-calibrated phylogeny and full mitogenome sequence of the Galapagos sea lion (*Zalophus wollebaeki*) from scat DNA

Pacarina Asadobay[1], Diego O. Urquía[1], Sven Künzel[2], Sebastian A. Espinoza-Ulloa[3,4], Miguel Vences[5] and Diego Páez-Rosas[1,6]

[1] Galapagos Science Center, Universidad San Francisco de Quito, Isla San Cristóbal, Islas Galápagos, Ecuador
[2] Department of Evolutionary Genetics, Max Planck Institute, Plön, Germany
[3] Facultad de Medicina, Pontificia Universidad Católica del Ecuador, Quito, Pichincha, Ecuador
[4] Department of Biology, University of Saskatchewan, Saskatoon, Canada
[5] Zoological Institute, Technische Universität Braunschweig, Braunschweig, Germany
[6] Oficina Técnica San Cristóbal, Dirección Parque Nacional Galápagos, Isla San Cristóbal, Islas Galápagos, Ecuador

## ABSTRACT

**Background**. The Galapagos sea lion, *Zalophus wollebaeki*, is an endemic and endangered otariid, which is considered as a sentinel species of ecosystem dynamics in the Galapagos archipelago. Mitochondrial DNA is an important tool in phylogenetic and population genetic inference. In this work we use Illumina sequencing to complement the mitogenomic resources for *Zalophus* genus—the other two species employed Sanger sequencing—by a complete mitochondrial genome and a molecular clock of this species, which is not present in any case.

**Materials and Methods**. We used DNA obtained from a fresh scat sample of a Galapagos sea lion and shotgun-sequenced it on the Illumina NextSeq platform. The obtained raw reads were processed using the GetOrganelle software to filter the mitochondrial *Zalophus* DNA reads (∼16% survive the filtration), assemble them, and set up a molecular clock.

**Results**. From the obtained 3,511,116 raw reads, we were able to assemble a full mitogenome of a length of 16,676 bp, consisting of 13 protein-coding genes (PCGs), 22 transfer RNAs (tRNA), and two ribosomal RNAs (rRNA). A time-calibrated phylogeny confirmed the phylogenetic position of *Z. wollebaeki* in a clade with *Z. californianus*, and *Z. japonicus*, and sister to *Z. californianus*; as well as establishing the divergence time for *Z. wollebaeki* 0.65 million years ago. Our study illustrates the possibility of seamlessly sequencing full mitochondrial genomes from fresh scat samples of marine mammals.

Corresponding author
Diego Páez-Rosas, dpaez@usfq.edu.ec

## INTRODUCTION

The family Otariidae is a main clade of pinnipeds (*Berta & Churchill, 2012*; *Berta, 2018*), including seven genera and 14 species. It is morphologically characterized by an auditory pinna, and posterior limbs directed forward (*Wursig, Perrin & Thewissen, 2009*; *Berta & Churchill, 2012*). The distribution of otariids comprises the coasts of North and South America, the coasts of North Asia, Australia, New Zealand, and in addition several island systems including the Galapagos archipelago (*Wursig, Perrin & Thewissen, 2009*). The Galapagos sea lion, *Zalophus wollebaeki* (*Sivertsen, 1953*), is an endemic and endangered otariid species (*Trillmich, 2015*), which concentrates its largest populations in the southeastern part of the archipelago (*Riofrío-Lazo, Arreguín-Sánchez & Páez-Rosas, 2017*; *Páez-Rosas et al., 2021*). Given its position as top predator and its high nutritional efficiency, it is considered as a sentinel species of the ecosystem (*Páez-Rosas & Guevara, 2017*), and its permanence in the region is assumed to regulate the health and functioning of the Galapagos marine ecosystem at all trophic levels (*Fariña et al., 2003*; *Riofrío-Lazo et al., 2021*). However, it remains exposed to periodic oceanographic-atmospheric events in the region, such as warm phases of the El Niño–Southern Oscillation, which has caused drastic population decreases in the last four decades (*Páez-Rosas et al., 2021*).

Despite the advent of high-throughput genomic approaches, mitochondrial DNA remains an important tool in phylogenetic and population genetic inference (*DeSalle & Hadrys, 2017*), and has also been used for population genetics of *Z. wollebaeki* (*Arnason et al., 2006*). In animals, due to its high copy numbers compared to nuclear DNA, it is a particularly suited marker to be obtained from low-quality DNA samples (*DeSalle & Hadrys, 2017*). For the genus *Zalophus*, complete mitogenome sequences have been published both for the California sea lion, *Z. californianus*, and the extinct Japanese sea lion, *Z. japonicus* (*Arnason et al., 2006*; *Kim et al., 2021*). Here, we complement the mitogenomic resources for *Zalophus* genus, by a complete mitogenome, sequenced from DNA extracted from a scat sample of *Z. wollebaeki*. In the same way, we constructed a molecular clock using the complete mitogenomes of otariid species.

## MATERIALS AND METHODS

A fresh scat sample was obtained from an *Zalophus wollebaeki* individual on the El Malecón rookery at San Cristóbal island, Galapagos archipelago (Lat: −0.90072, Long: −89.610117) in July 2021 as part of a trophic ecology project. The sample was stored in the Molecular Biology Laboratory of the Galapagos Science Center (GSC) in a −20 °C freezer. Total DNA was extracted with the QIAamp Fast DNA Stool Mini Kit (Qiagen, Hilden, Germany) from 220 mg of sample following manufacturer instructions with some modifications such as decreased incubation temperature from 70 to 50 °C, and the final elution of extracted DNA in 100 µl of TAE buffer. Agarose gel electrophoresis (1%) was performed to check purity and integrity of DNA, while a NanoDrop 2000 spectrophotometer was used to check quantity, yielding 56.1 ng/µl of DNA.

A library of the total extracted DNA was prepared using an Illumina TruSeq stranded protocol and sequenced together with 12 unrelated (RNAseq) samples on an Illumina

NextSeq system using a 2 × 150 cycles using a High Output kit. The obtained raw reads (3,511,116) were processed using the GetOrganelle software (*Jin et al., 2020*) to filter the mitochondrial *Zalophus* DNA reads (remaining ∼16% of raw reads) and assemble them. GetOrganelle was chosen because it deals appropriately with contamination, an evident issue with the scat sample we work with (*Freudenthal et al., 2019*). Stool contains DNA from other sources, including mainly bacteria but also some prey, parasites, and environmental contaminants. GetOrganelle firstly filters read mapping to a mitochondrion reference, discarding most of the contaminant reads in this step (*Jin et al., 2020*). Then, it employs the SPAdes assembler that is designed for single-cell sequencing, so it allows to discriminate sequences from different sources with different coverages (*Bankevich et al., 2012*). GetOrganelle then recognizes the target mitogenome based on connections, coverages, and BLAST hits, putting aside any remaining contamination sequences (*Jin et al., 2020*).

In order to adjust the reading frame of the obtained assembly and to facilitate its alignment with published mitogenomes, its reverse complement was obtained, and the circularization cut-off was set in the position 373; in this way we succeeded obtaining a final quasi-mitogenome assembly for *Z. wollebaeki*. Quality control of the final mitogenome assembly was assessed by getting a complete annotation of it using the MITOS Web server 2.0 (*Donath et al., 2019*), and by comparing the nucleotide similarity (*via* NCBI Megablast) of this mitogenome with the references of *Z. japonicus*, *Z. californianus* and *Eumetopias jubatus* (Steller's sea lion). Syntheny across the three *Zalophus* species was assessed by comparing their mitogenomes' annotations; a codon usage analysis of the three species was also performed employing the Sequence Manipulation Suite (*Stothard, 2000*). Finally, the *Z. wollebaeki* mitogenome map was constructed using OGDRAW 1.3.1 (*Greiner, Lehwark & Bock, 2019*).

A time-calibrated phylogenetic tree was constructed including the newly assembled *Z. wollebaeki* mitogenome together with other complete otariid mitogenomes published at GenBank (Table 1). The harbor seal (*Phoca vitulina*), grey seal (*Halichoerus grypus*), and walrus (*Odobenus rosmarus*) mitogenomes were set as outgroups. All the mitogenomes were aligned in MAFFT online (*Katoh, Rozewicki & Yamada, 2019*). Then, JModelTest 2.1.10 (*Darriba et al., 2012*) was employed to choose the most appropriate evolutionary model, using all the best model selection tests (*i.e.* AIC, BIC and DT). All tests indicated that the best model for the data set was GTR+I+G. With the information from JModelTest it was possible to adjust the initial parameters for Gamma, portion of invariant sites, nucleotide frequencies and nucleotide exchange rates that were included in the subsequent Bayesian analysis to determine divergence ages. MrBayes 3.2.7a (*Ronquist et al., 2012*) was used for Bayesian and coalescent analysis, fitting the parameters retrieved from JModelTest, and including the calibrations (priors) from Table 2. A MCMC of 3,000,000 generations were run (with a burn-in of 25%). Sampling was taken every 100 records and the molecular clock was set to strict. The resulting parameters of interest showed outstanding convergence. The time scale was adjusted in the tree using 2% divergence. The resulting tree was visualized in FigTree version 1.4.4 (*Rambaut, 2010*).
**Table 1   Complete mitogenome of *Zalophus wollebaeki*.** Taxa with complete mitogenome sequences used for the phylogenetic analyses.

| Accession number | Species | Sequence length (bp) |
|---|---|---|
| OP115880 | *Zalophus wollebaeki* | 16,676 |
| NC_008416_1 | *Zalophus californianus* | 16,677 |
| NC_058016_1 | *Zalophus japonicus* | 16,698 |
| NC_004030_2 | *Eumetopias jubatus* | 16,638 |
| NC_049152_1 | *Otaria byronia* | 16,643 |
| NC_008417_1 | *Arctocephalus pusillus* | 16,652 |
| NC_004023_1 | *Arctocephalus forsteri* | 15,413 |
| NC_008420_1 | *Arctocephalus townsendi* | 16,571 |
| NC_008419_1 | *Neophoca cinerea* | 16,736 |
| NC_008418_1 | *Phocarctos hookeri* | 16,722 |
| NC_008415_3 | *Callorhinus ursinus* | 16,668 |
| NC_001325_1 | *Phoca vitulina* | 16,826 |
| NC_004029_2 | *Odobenus rosmarus* | 16,565 |
| CM022781_1 | *Halichoerus grypus* | 16,773 |

**Table 2   Divergence estimation priors set for time-calibrated phylogenetic tree construction in Mr Bayes.** Median time estimation is presented in millions of years (mya).

| Diverging groups | Median time estimation (95% quantile) | Fossil estimation was based on | Reference |
|---|---|---|---|
| Otariidae/rest of pinnipeds | 16.0 mya (15.20–16.80) | *Eotaria crypta* and *Eotaria citrica* | *Boessenecker & Churchill (2015)*; *Velez-Juarbe (2017)* |
| *Arctocephalus* (*Arctophoca*)/rest of otariids | 6.05 mya (5.39–6.79) | *Hydrarctos lomasiensis* | *Muizon (1978)*; *Churchill, Boessenecker & Clementz (2014)* |
| *Phoca vitulina*/*Halichoerus grypus* | 4.48 mya (4.40–4.56) | Basal 'Phocina' split | *Arnason et al. (2006)*; *Wolf, Tautz & Trillmich (2007)* |
| Pinnipeds root | 25.7 mya (23.40–28.10) | *Enaliarctos tedfordi* | *Berta (1991)* |

## RESULTS

We obtained a final *Zalophus wollebaeki* mitogenome assembly of 16,676 bp in length (GenBank: OP115880), with a GC content of 40.5%. The average coverage of the mitogenome was 55×. The newly assembled *Z. wollebaeki* mitogenome consists of 13 protein-coding genes (PCGs), 22 transfer RNAs (tRNA), and two ribosomal RNAs (rRNA) (Fig. 1), agreeing that seen in the other *Zalophus* mitogenomes. Some gaps (labelled as Ns in our mitogenome) were observed in the non-coding control region (D-loop) when aligning the *Z. wollebaeki* mitogenome with the published ones, these being potential segments that couldn't be successfully assembled. However, the total length of these gaps did not exceed the 100 bp total.

Sequence comparison revealed that *Z. californianus* was 99.42% similar to our *Z. wollebaeki* mitogenome (16,346/16,439 bp were identical), with only one gap (1 bp indel)

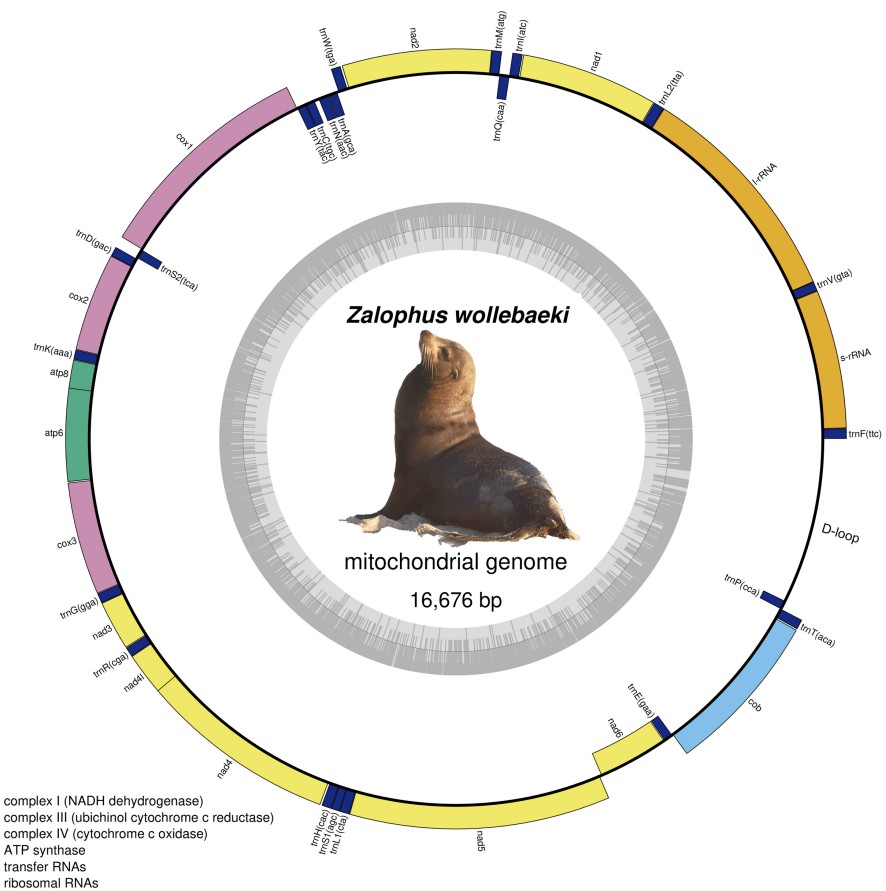

**Figure 1** **Mitogenome of *Zalophus wollebaeki*.** Gray bars indicate the GC content of each section.

found. In the same way, *Z. japonicus* showed 98.65% of similarity (16,215/16,440 bp were identical), with four gaps found. On the other hand, the *Eumetopias jubatus* mitogenome displayed a similarity of 94.11% with respect to *Z. wollebaeki*.

The codifying elements at the mitogenomes of the three *Zalophus* species were perfectly collinear and no major indels, inversions or translocations were found when comparing them in synteny analysis. Codon usage analysis revealed that *Z. wollebaeki* and *Z. californianus* are biased towards the same codon at 16/20 aminoacids plus the stop codon. When comparing *Z. wollebaeki* and *Z. japonicus*, they are biased for the same codon at only 13/20 aminoacids plus the stop codon (Table 3).

The time-calibrated phylogenetic tree (Fig. 2) showed a monophyletic group containing the three species of *Zalophus*, supporting the sister group relationship of *Z. californianus* and *Z. wollebaeki* with a divergence time estimation of 0.65 mya. It also shows the genus *Zalophus* diverged from *E. jubatus* 7.98 mya. Our phylogeny supports *Eumetopias* being sister to the genus *Zalophus*, while the clade containing these two genera would be sister to the remaining otariids (Southern hemisphere otariids). The divergence times confidence

**Table 3  Codons for the *Zalophus* species.** The most frequently used codon for each aminoacid in each species is shown, as well as its frequency (in brackets). It is highlighted in bold when the most frequent codons are the same between *Zalophus* species.

| Aminoacid | *Zalophus wollebaeki* | *Zalophus californianus* | *Zalophus japonicus* |
|---|---|---|---|
| Ala | **GCC (0.38)** | **GCC (0.37)** | GCA (0.35) |
| Cys | **TGC (0.72)** | **TGC (0.58)** | TGT (0.52) |
| Asp | **GAC (0.53)** | **GAC (0.54)** | **GAC (0.55)** |
| Glu | **GAA (0.63)** | **GAA (0.59)** | GAA (0.62) |
| Phe | **TTC (0.59)** | **TTC (0.52)** | TTC & TTT (0.50) |
| Gly | **GGA (0.38)** | **GGA (0.39)** | **GGA (0.36)** |
| His | CAC (0.53) | CAT (0.55) | CAT (0.53) |
| **Ile** | **ATT (0.52)** | **ATT (0.53)** | ATC (0.52) |
| **Lys** | **AAA (0.73)** | **AAA (0.78)** | **AAA (0.70)** |
| **Leu** | **CTA (0.36)** | **CTA (0.34)** | **CTA (0.27)** |
| **Met** | **ATA (0.74)** | **ATA (0.69)** | **ATA (0.65)** |
| **Asn** | **AAT (0.53)** | **AAT (0.53)** | **AAT (0.54)** |
| Pro | CCA (0.34) | CCT (0.36) | CCT (0.34) |
| Gln | **CAA (0.68)** | **CAA (0.70)** | **CAA (0.67)** |
| Arg | CGA (0.30) | CGT (0.30) | **CGA** & CGT (0.29) |
| Ser | **TCA (0.27)** | **TCA (0.26)** | **TCA (0.28)** |
| Thr | **ACA (0.36)** | **ACA (0.32)** | **ACA (0.31)** |
| Val | **GTA (0.44)** | **GTA (0.47)** | **GTA (0.43)** |
| Trp | **TGA (0.60)** | **TGA (0.67)** | **TGA (0.73)** |
| Tyr | TAC (0.52) | TAT (0.55) | TAT (0.54) |
| STOP | **TAA (0.50)** | **TAA (0.38)** | **TAA (0.44)** |

intervals are mostly quite tight, demonstrating the high degree of convergence of the substitution rates and thus of the node ages.

## DISCUSSION

The general features of *Zalophus wollebaeki* mitogenome—such as length, GC content, and PCGs, rRNAs, and tRNAs found—were consistent to those of the other two *Zalophus* mitogenomes: *Z. japonicus* (16,698 bp; *Kim et al., 2021*) and *Z. californianus* (16,677 bp; *Arnason et al., 2006*). It must be noted that *Z. wollebaeki* was sequenced with the Illumina method, while *Z. japonicus* and *Z. californianus* were sequenced through Sanger (*Arnason et al., 2006*; *Kim et al., 2021*). All three sequences presented a high similitude regardless of the sequencing method used. Confirming that any type of sequencing can recovered a complete mitogenome.

Our phylogenetic reconstruction supports the genus *Zalophus* being monophyletic, with *Z. californianus* being the sister species and hence a possible antecessor of *Z. wollebaeki* (*Wolf, Tautz & Trillmich, 2007*; *Wolf et al., 2008*). The perfect collinearity and high sequence similarity (>98%) across these three species further support monophyly in *Zalophus*. Although a recent phylogenomics study failed in supporting the monophyly of
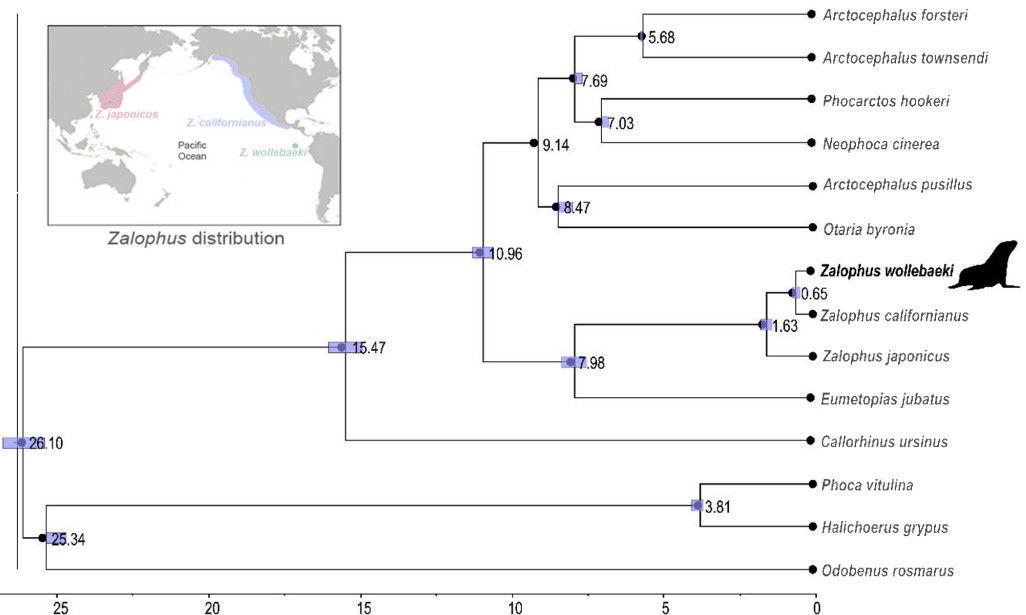

**Figure 2 Time calibrated phylogeny of *Zalophus wollebaeki*.** Time calibrated phylogeny based on 11 mitogenomes sequences of Otariidae plus 3 pinniped outgroups. Nodes are labelled with their respective divergence time estimations (millions of years ago) and the bar represents the confidence intervals of these estimates. The *Zalophus wollebaeki* mitogenome obtained in this research is highlighted in bold.

the *Zalophus* genus (*Lopes et al., 2021*), most of the body of evidence in otariid phylogeny supports consistently this pattern (*e.g.*, *Yonezawa, Kohno & Hasegawa, 2009*; *Churchill, Boessenecker & Clementz, 2014*). Our results support that *Eumetopias jubatus* is sister to the genus *Zalophus*, while *Eumetopias* and *Zalophus* together would be sister to the remaining otariids. This also coincides with other phylogenetic studies in Otariids, from those based on mitochondrial sequences only (*Yonezawa, Kohno & Hasegawa, 2009*; *Churchill, Boessenecker & Clementz, 2014*; *Austin, Eriksen & Bachmann, 2023*) to those based on phylogenomics (*Lopes et al., 2021*).

Our phylogeny, high nucleotide similarity and close codon usage patterns between *Z. wollebaeki* and *Z. californianus* supports your relationship of sister species (*Athey et al., 2017*). However, enough nucleotide differences (including 16 substitutions at the D-loop region) would exist for supporting *Z. wollebaeki* being a separated species from *Z. californianus*, as also suggested from the *Z. wollebaeki* paratype mitogenome analysis (*Austin, Eriksen & Bachmann, 2023*). According to our estimations, the *Z. wollebaeki* and *Z. californianus* split would be dated 0.65 mya, a more recent date than the 2.5 mya estimated by *Wolf, Tautz & Trillmich (2007)*. Such a recent divergence between both species agrees with the evident morphological similarity they share, being size the only distinguishable difference between them (*Z. wollebaeki* is smaller) (*Sivertsen, 1953*; *Orr & Scheffer, 1958*); reason why until recently, *Z. wollebaeki* was classified as a subspecies of *Z. californianus* (*Wolf, Kauermann & Trillmich, 2005*; *Wolf, Tautz & Trillmich, 2007*). Our

estimation of 0.65 mya also matches with the Kansas glaciation, an especially strong glacial period occurred 0.7−0.6 mya (*Aber, 1991*). Global cooling events as this one increased ocean productivity even in equatorial waters, facilitating the spread of pinnipeds across hemispheres and tropical environments (*Churchill, Boessenecker & Clementz, 2014*; *Lopes et al., 2021*). Therefore, the arrival of the ancestors of *Z. wollebaeki* into the Galapagos Islands and its divergence from *Z. californianus* around this time, is a perfectly plausible event.

We successfully annotated in *Z. wollebaeki* mitogenome all the genes expected in every mammal mitogenome. Thus, our study demonstrates the prospect of recovering complete and accurate mitogenomes of the host from scat samples in the field. A similar method has previously been used for mitogenome sequencing of mammals, including artiodactyls (*i.e.*, bharal *Pseudois schaeferi*) (*Dong et al., 2019*), and primates (*i.e.*, toque macaque *Macaca sinica*) (*Roos, 2018*). In the latter case, DNA extracted from feces yielded complete mitogenomes by Sanger sequencing, with sufficient coverage to assemble it without inconvenience, leaving aside the rest of reads present in the feces (*Roos, 2018*). Note we achieved highly supported results, with accurate time estimates and no phylogenetic ambiguities, by analyzing entire mitogenomes without partitioning them for modeling. Thus, our work joins other phylogenetic studies among closely related species (*e.g.*, *Zhang et al., 2021*; *Zawal et al., 2022*; *Sun et al., 2023*) where the use of whole mitogenomes instead of partitioning yields better results by preventing the addition of unnecessary complexity and bias in our analyses, while providing a greater amount of information for inference (*Duchêne et al., 2011*).

Despite the unspecific approach used herein (shotgun sequencing of total DNA, which mostly yielded DNA reads of bacteria), assembly was straightforward and did not require any filtering approach. This suggests that host mitochondrial DNA was common in the DNA extracted from the scat sample, and contaminants from other metazoans did not influence the assembly in a meaningful way. The assembly process in our study certainly was also facilitated by the existence of mitogenome sequences of related species of *Zalophus*. We obtained, with a fecal DNA extraction kit, a DNA concentration of 56.1 ng/μl which was sufficient for library preparation and assembly of a sequence of 16,676 bp, but we assume that with less fresh samples, DNA yield and proportion of predator mtDNA may be less favorable. So, we expect the novel method presented in this work to be applicable for fresh scat samples of other pinniped species.

## CONCLUSIONS

We stress the novelty of our study, obtaining a complete mitogenome (excepting by a <100 bp stretch) for *Zalophus wollebaeki* from fresh scats collected in the field, in a non-invasive sampling. With this information we were able to compare distinct sequences and estimate the divergence time of *Z. wollebaeki,* which matched with a global cooling event. Additionally, we supported the sister-species relationship of *Z. wollebaeki* and *Z. californianus*, as well as the monophyly of the *Zalophus* genus.

## ACKNOWLEDGEMENTS

We thank the staff of the Galapagos Science Center (GSC), and the Zoological Institute of the Technische Universität Braunschweig (TUBS) for their logistical, and technical support. Finally, we thank the anonymous reviewers of this manuscript for their great contributions.

### Funding

This work was supported by the USFQ Collaboration Grant and Galapagos Science Center Grant (POA 2021). The funders had no role in study design, data collection and analysis, decision to publish, or preparation of the manuscript.

### Grant Disclosures

The following grant information was disclosed by the authors:
USFQ Collaboration Grant and Galapagos Science Center: POA 2021.

### Competing Interests

The authors declare there are no competing interests.

### Author Contributions

- Pacarina Asadobay performed the experiments, analyzed the data, prepared figures and/or tables, authored or reviewed drafts of the article, and approved the final draft.
- Diego O. Urquía conceived and designed the experiments, performed the experiments, analyzed the data, prepared figures and/or tables, authored or reviewed drafts of the article, and approved the final draft.
- Sven Künzel performed the experiments, authored or reviewed drafts of the article, and approved the final draft.
- Sebastian A. Espinoza-Ulloa analyzed the data, prepared figures and/or tables, authored or reviewed drafts of the article, and approved the final draft.
- Miguel Vences conceived and designed the experiments, performed the experiments, authored or reviewed drafts of the article, and approved the final draft.
- Diego Páez-Rosas conceived and designed the experiments, authored or reviewed drafts of the article, and approved the final draft.

### Field Study Permissions

The following information was supplied relating to field study approvals (i.e., approving body and any reference numbers):

The Galapagos National Park Directorate (DPNG) and the Ecuadorian Ministry of the Environment for granting us the research permissions (No. PC-31-21-003; MAATE-DBI-CM-2021-0178) provided permission to carry on this study.

## Data Availability

The mitogenome sequence data are available at NCBI GenBank: OP115880, PRJNA892743, SAMN31360129 and SRR22026703.

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
