# Peer review of "Time-calibrated phylogeny and full mitogenome sequence of the Galapagos sea lion (Zalophus wollebaeki) from scat DNA"

_PeerJ, doi:10.7717/peerj.16047_

## Round 0.1 · original submission · Major Revisions

Dear Authors,

After receiving feedback from three reviewers, it has been determined that your study presents an interesting and novel approach. However, two of the reviewers have raised concerns regarding the level of detail provided in certain aspects of the methodology, and have suggested the need for additional analyses to corroborate the findings from BEAST. Additionally, some other observations have been made about the text, including the need to improve the discussions in order to better highlight the importance of the results. Significant revisions are therefore necessary before the paper can be accepted for publication.

Sincerely,

Armando Sunny.

Reviewer 1 ·

Basic reporting

The manuscript entitled “The complete mitogenome of the Galapagos sea lion
(Zalophus wollebaeki) from scat DNA” accomplish the basic reporting criteria in terms of writing, structure and relevance.

Experimental design

I want to focus here in two methodological concerns:

-A partition-based analysis (eg. Model Finder, Partition finder) of phylogenetic position of Zalophus wollebaeki mitochondial genome may be more appropiate, instead the concatenated analyses.

-I wondering if the authors also tested a Yule prior model tree apart of coalescent model for their BEAST reconstruction and, if that was the case, the estimated divergence times resulted similar in both models.

Please check “additional comments” for specific annotations and questions about each section

Validity of the findings

Analyses pipeline are concise and enought for exploratory puposes of the new mitochondrial genome isolated and assembled of Z. wollebaeki.

Additional comments

I’m including here particular comments for all sections in the ms:

Line 50: change extremities for limbs
Line 113: Maybe the authors refer here to JModelTest instead just JModel (Posada D. jModelTest: phylogenetic model averaging. Mol Biol Evol. 2008 Jul;25(7):1253-6. doi: 10.1093/molbev/msn083)
Line 115: Just GTR or GTR + i + G ?
Line 120: Change generations instead repetitions
Line 130: The authors may consider also to include codon usage/percentage and ATGC composition as a descriptive features in their inferance
Figure 2: Please, annotate the divergence times numbers bigger and clearer (maybe keeping just two decimal places instead four) and hide the branch supports, in order to facilitate the visualisation of divergence times.
The authors may consider include a geographic distribution representation of Zalophus members, probably with a small map within BEAST chronogram, a completly new figure appart is not necesary.

Reviewer 2 ·

Basic reporting

Unfortunately, my English is not native, so I did not do a very deep critique of writing and grammar.

The references used in the article are fine, however the authors can enrich the discussion of their results by incorporating more relevant references. For example, in the discussion of the successful use of fecal material there is a paragraph that does not contain any citations to support their claims. In another section they cite articles from 16 years ago and more, arguing that it is something recent. Authors are suggested to search deeper into the topics discussed to better exploit their results and have adequate references to support what is written.

Regarding the structure of the article, the results are very scarce and sometimes quotes are included as if it were the discussion section. While in discussion they do the opposite. It is suggested to restructure your results and discussion, making it clear that the results are only from your work and it is in the discussion where support is given to these results and justifications in cases where a new finding is found.

Experimental design

It is not clear what is the relevance of using the complete mitogenome or that this study results beyond the divergence times. If the only relevant result is the estimate of the divergence times, the title should be different.

The authors mention in the title and the introduction the relevance of complementing the mitochondrial genomes of the genus Zalophus with the assembled mitogenome, however throughout the article the characteristics of this mitogenome are not addressed in depth. It is suggested to justify why deeper analyzes on the characteristics of the mitogenome were not carried out or, otherwise, to implement more descriptive analyzes that allow understanding the particularities of this new mitogenome compared to the congener species that were available in Genbank. Especially in the case of uncertainty about whether they are sister species or subspecies.

In the methods, the use of BEAST to estimate the divergence times is not clear, the methods used and the parameters with which the analyzes were run should be clearly clarified.

The names of the programs and the version that is used is not well written in all cases. This information must be included so that the analyzes can be replicated by other scientists.

Throughout the text the authors mention that they assembled the complete mitochondrial genome, however in the conclusion they mention it as quasi-complete, they should clarify why this is mentioned only until the end.

Validity of the findings

The analyzes presented are insufficient to validate the importance of assembling this mitogenome for the genus Zalophus. It is suggested to analyze this mitogenome in more detail and better exploit the information generated.

The main conclusion is not well supported in the discussion and everything that is discussed in this section is not mentioned in the conclusion. It is suggested that when restructuring the results and discussion section, a conclusion more in line with the real contributions of the work is drawn up.

Additional comments

Keep in mind that the first time the scientific name of the species is mentioned in each section, the complete genus must be written. After the first mention you can already abbreviate the genre.

Abbreviations that have not been described in the figure caption should not be added to tables and figures.

Only the elements contained in the figure should be described in the legend that accompanies figure 1. "Other genes" are not present in the figure, therefore they should not be included in the legend.

See more detailed comments on the article in the attached file.

Annotated reviews are not available for download in order to protect the identity of reviewers who chose to remain anonymous.

Reviewer 3 ·

Basic reporting

This manuscript is about the assemblage of complete mitochondrial genome obtained from the scat of Galapagos sea lion (Zalophus wollebaeki). This species is known as the endangered species and its genetic information would be valuable for various purposes. However, we found the previous report about its genome sequence from Genbank database (NC_062331.1). This result may weaken the originality and novelty of authors studies. Additionally, DNA in the scat is highly degraded and sequence accuracy in the control region should be reexamined using long read rather than short Illumina sequencer. Therefore, I am afraid that the novelty and accuracy of the mitogenome sequence should be addressed before further review.

Experimental design

experimental design is generally accepted method. however, they should confirm the control region using the sanger sequence again.

Validity of the findings

it is worth to secure of genetic resources of endangered species such as Galapagos Sea lion. However, there is previous report about the same species. For further consideration, I would like to recommend to conduct study about biogeographical difference with additional samples.

---

## Round 0.2 · Minor Revisions

DDear Authors,

I am delighted to share some positive news with you. Two out of the three reviewers have expressed their satisfaction with the corrections made to the manuscript. However, they have kindly suggested a few minor adjustments, which, once implemented, will lead to the acceptance of your work for publication. Rest assured, I will expedite the process once these last changes are completed.

Thank you for considering PeerJ as the potential venue for publishing your intriguing manuscript.

Best regards,

Armando Sunny

Reviewer 1 ·

Basic reporting

Since I’m not a native speaker, I can only confirm that references and general structure of the manuscript are clear an sufficient for the required purposes.

Experimental design

Sampling, laboratory and bioinformatic procedures are reasonably well done

Validity of the findings

Overall findings and conclusions are clearly stated. Conclusions and insights of this work would result of interest in specific fields of zoology, phylogenetics and bioinformatics.

Additional comments

I consider all comments and suggestions from the first revision round were successfully considered and this version has been considerably enhanced with respect the previous one. The authors answered with respect partition vs. concatenated analysis question the following comment: “Additionally, it is important to mention that the use of whole mitogenomes instead of partitioning has become very common and, in some cases, mandatory in recent studies”. I wonder if authors may add a brief comment and citation(s) within discussion section about this approach and explicitly suggest in which cases concatenated analyses are recommended or less biased with respect partitioned approaches.

Reviewer 2 ·

Basic reporting

In its current version, the manuscript meets the basic reporting criteria in terms of writing, structure, and relevance.

Experimental design

The analyzes implemented to explain its results seem more appropriate to this version.

Validity of the findings

By restructuring their results and discussion, the authors manage to highlight the importance of their work in a more adequate, clear and concise way.

Additional comments

Line 280: The reference is not cited in the text, correct this or delete it.

Line 360-361: In figure 2 all the names of the species are highlighted in bold, change the wording to allude to the mark that accompanies the name of the species assembled in this work.

Reviewer 3 ·

Basic reporting

All the answers that authours reply to my questions were below my expectations.
First, I asked if there is novel findings about this paper, since its mitogemone sequence was already published from the other group. I understand that this authors submitted one month prior to those who published. However, it does not mean that this manuscript has novelty.
Second, I asked the confirm the D-loop with Sanger method and they did not.
Although they added several bioinformation data, which can be easily produced.
Therefore, I think this manuscript was not improved enough for publication.

Experimental design

overall experimental design was acceptable

Validity of the findings

As I mentioned above, this manuscript lacks novelty.

---

## Round 0.3 · accepted · Accept

Dear authors,

It is my pleasure to inform you that the manuscript has been accepted for publication since both reviewers had commented that after making minor corrections it could already be accepted. Thank you very much for choosing PeerJ for the publication of such an interesting manuscript.

Sincerely,

Armando Sunny